EMBO
Molecular Medicine

# A complex genomic locus drives mtDNA replicase POLG expression to its disease-related nervous system regions

Joni Nikkanen[1], Juan Cruz Landoni[1], Diego Balboa[1,2], Maarja Haugas[3], Juha Partanen[3], Anders Paetau[4], Pirjo Isohanni[1,5], Virginia Brilhante[1] & Anu Suomalainen[1,6,7,*]

## Abstract

DNA polymerase gamma (POLG), the mtDNA replicase, is a common cause of mitochondrial neurodegeneration. Why POLG defects especially cause central nervous system (CNS) diseases is unknown. We discovered a complex genomic regulatory locus for *POLG*, containing three functional CNS-specific enhancers that drive expression specifically in oculomotor complex and sensory interneurons of the spinal cord, completely overlapping with the regions showing neuronal death in POLG patients. The regulatory locus also expresses two functional RNAs, *LINC00925*-RNA and MIR9-3, which are coexpressed with *POLG*. The MIR9-3 targets include NR2E1, a transcription factor maintaining neural stem cells in undifferentiated state, and MTHFD2, the regulatory enzyme of mitochondrial folate cycle, linking POLG expression to stem cell differentiation and folate metabolism. Our evidence suggests that distant genomic non-coding regions contribute to regulation of genes encoding mitochondrial proteins. Such genomic arrangement of *POLG* locus, driving expression to CNS regions affected in POLG patients, presents a potential mechanism for CNS-specific manifestations in POLG disease.

**Keywords** enhancer; gene regulation; mtDNA maintenance; POLG; tissue specificity

**Subject Categories** Chromatin, Epigenetics, Genomics & Functional Genomics; Genetics, Gene Therapy & Genetic Disease; Neuroscience

## Introduction

Mutations in the genes encoding mitochondrial DNA (mtDNA) replisome proteins cause mtDNA maintenance defects, which lead to common metabolic disorders (Viscomi & Zeviani, 2017). The most common nuclear gene underlying mitochondrial disorders is the mtDNA replicative polymerase gamma (POLG), with more than 145 disease mutations (http://tools.niehs.nih.gov/polg/). POLG is a prime example of tissue- and genotype-specific variability of manifestations, and POLG mutations cause highly tissue-specific disorders typically manifesting in the nervous system, but also affecting the liver and skeletal muscle. Despite the apparent requirement of mtDNA replication in cell division, the patients with POLG disorders do not typically show symptoms deriving from highly proliferating cell types, such as anemia. The reasons for such postmitotic cell manifestations—and mechanisms of tissue specificity overall—are unknown.

The clinical manifestations of the mutations of POLG are diverse: (i) Alpers–Huttenlocher syndrome, an epileptic encephalohepatopathy of early childhood, manifesting during the first years of life and typically progressing to death within a few years (Naviaux & Nguyen, 2004); (ii) teenage-onset epileptic encephalopathy (SCA-E; Winterthun et al, 2005); (iii) adult-onset mitochondrial recessive ataxia syndrome (MIRAS) (Hakonen et al, 2005; Winterthun et al, 2005); and (iv) adult-onset progressive external ophthalmoplegia (PEO) with or without sensory neuropathy, often associated with parkinsonism and premature menopause (Van Goethem et al, 2001; Luoma et al, 2004). Epilepsy in POLG syndrome may develop to status epilepticus and is associated with poor prognosis (Hikmat et al, 2017). The patients may develop cognitive defects and complex psychiatric manifestations, ranging from avoidant personality to depression or paranoia (Hakonen et al, 2005). The mechanistic basis of the exceptionally variable nervous system defects of POLG—from epilepsy to cognitive decline, ataxia, psychiatric symptoms, and parkinsonism—is unknown. Furthermore, even patients with the same ancestral MIRAS allele (homozygous allelic mutations leading to p.W748S+E1143Q amino acid changes) show variable disease manifestations, with SCA-E, MIRAS, or

1 Research Programs Unit, Molecular Neurology, University of Helsinki, Helsinki, Finland
2 Biomedicum Stem Cell Center, University of Helsinki, Helsinki, Finland
3 Department of Biosciences, University of Helsinki, Helsinki, Finland
4 HUSLAB and Department of Pathology, University of Helsinki and Helsinki University Hospital, Helsinki, Finland
5 Department of Pediatric Neurology, Children's Hospital, University of Helsinki and Helsinki University Hospital, Helsinki, Finland
6 Department of Neurology, University of Helsinki and Helsinki University Hospital, Helsinki, Finland
7 Neuroscience Center, University of Helsinki, Helsinki, Finland
*Corresponding author. Tel: +358 9 4717 1965; E-mail: anu.wartiovaara@helsinki.fi

PEO–polyneuropathy–parkinsonism (Hakonen *et al*, 2005). The variability of CNS manifestations even in patients with the same disease allele indicates strong modifier effects/genes contributing to the manifestations. These observations prompted us to explore the genomic regions of mtDNA maintenance loci to identify potential genomic modifiers for these disorders.

## Results and Discussion

To clarify the upstream regulation of the mtDNA replicase expression, we analyzed the activity of the proximal promoter of *POLG*. Its promoter has been reported to drive relatively low expression levels in different tissues and organisms (Carninci *et al*, 2005). However, in a luciferase expression system *in vitro*, the promoter was highly active, comparable to viral SV40 promoter (Fig 1A), and required the presence of one of the two predicted CAAT boxes (Fig 1A and B). When expressed *in vivo*, however, in transgenic E12.5 mouse embryos, the *Polg* promoter was active especially in the midbrain, dorsal root ganglia (DRG), developing motoneurons of the neural tube, and in skeletal muscle somites with very low expression outside CNS (Fig 1C). This muscle–CNS expression pattern was surprising for a mtDNA replicase, because of ubiquitous requirement of the mitochondrial genome replication, and raised the question whether the tissue-specific expression pattern of the proximal *Polg* promoter was modified by potential enhancer elements (EEs).

We utilized *in silico* prediction by Enhancer Element Locator program, which searches for conserved DNA elements of < 2,000 base pairs, which also show a conserved order of transcription factor binding sites within the EE (Hallikas *et al*, 2006). Within 100 kb up- and downstream of human, mouse, and rat *POLG* locus, we identified three strongly conserved putative EEs with high scores [768, 671, 522; when > 500 indicates likely enhancer (Hallikas *et al*, 2006)], 34–55 kb upstream of the coding region (Fig 1D, Appendix Table S1). Similar analysis for other mtDNA maintenance genes (*POLG2*, *TWNK*, *SSBP1*, and *TFAM*) suggested no strong candidates as their specific distant genomic regulators (Fig 1D).

To examine whether the three predicted EEs of the *POLG* locus were active *in vivo* in mice, we generated EE-specific transgenic mice. We cloned each of the mouse EEs separately in front of an HSV-tk minimal promoter driving *lacZ* expression (Hallikas *et al*, 2006). All the predicted EEs were biologically active and drove strong expression in distinct regions of the developing CNS of E12.5 embryos. No expression was detected in the liver or other organs (Fig 1E–M, Appendix Fig S1A). EE1 was active in proliferating immature neuronal precursors of the ventral and mid-trunk dorsal neural tube, EE2 and EE3 in dorsal neural tube, and EE2 also in DRG (Fig 1E–J). All the EEs also drove expression in specific brain regions (Fig 1K–M): EE1 expression overlapped with ISL1/2 motor neuron progenitors of the oculomotor complex, which innervate the extraocular and upper eyelid muscles (Fig 1K and N); EE2 in the superficial stratum of the superior colliculi (Fig 1L); EE3 in the dorsolateral midbrain, including both the ventricular and mantle zones (Fig 1M). None showed activity in somites. These results indicate that all EEs of the *POLG* locus are functional, nervous system-specific enhancers, with high specificity for defined neuronal regions.

Enhancers typically show different temporal characteristics, and therefore, we studied whether the three *Polg* EEs were also

functional in the adult brain. EE2 showed prominent expression in all of the EE2 transgenic lines with activity in the gray matter of the brain, most intensively in the hippocampus (CA1 and dentate gyrus > CA2 and 3), cortex, thalamus, mitral cell/external plexiform layer of olfactory bulb, cerebellar Purkinje, and granular cell layers (Fig 2A). Expression was also detected in the neural precursor regions: subventricular zone and rostral migratory stream (Fig 2B), which have been specifically found to be affected in POLG-deficient, progeric mtDNA mutator mice (Trifunovic *et al*, 2004; Ahlqvist *et al*, 2012). EE1 was not active in adults, and the expression patterns in different EE3 lines in the brain were inconclusive because of variability between transgenic lines. The results indicate that EE2 is the main CNS enhancer of *Polg* locus in adult mice, driving expression to the large neurons of neocortex and cerebellum as well as neural precursors.

In the adult mouse spinal cord, EE2 and EE3 showed overlapping, specific expression patterns in the laminae I-III of dorsal horns (Fig 2C–E) and the neuronal precursors of the central canal, which also were positive for POLG protein (Fig 2C and E). The dorsal horn sensory tracts receive nociceptive information from primary afferent nerves and contain mostly GABAergic, glycinergic, or glutamatergic interneurons (Todd, 2010). The EE2- and EE3-positive cells were often calbindin-positive (Fig 2F and G), pointing to activity in glutamatergic spinal interneurons. The evidence shows that EEs of the *Polg* locus drive expression to adult spinal sensory tracts and neural precursors of the spinal cord.

DNase I hypersensitive sites (DHSs) mark open DNA structures, typical for active promoters and regulatory elements (Gross & Garrard, 1988; Thurman *et al*, 2012). Furthermore, simultaneous opening of genomic elements on a chromosomal region is a strong indicator of functional cooperation of the two regions (Thurman *et al*, 2012). To clarify the genomic regions potentially regulating mtDNA maintenance genes, we analyzed their DHS patterns and correlated the DNase I sensitivity of distal DHSs with those proximal to the transcription start site of the target gene (Thurman *et al*, 2012). We found 123 distal DHSs correlating with *POLG* promoter DHSs (> 0.85 correlation), standing out from other mtDNA maintenance genes [*TWNK* (0), *POLG2* (0), *SSBP1* (1), and *TFAM* (5); Fig 3A and B]. These distant DHSs of the *POLG* region overlapped with the genomic locus of the enhancer elements, but also contain a long non-coding RNA gene (human *LINC00925*; mouse *Ai854517*; Fig 3C–E). Other genes in the *POLG* locus were found to have few DHSs *FANCI* (2), *RHCG* (0), and *TICRR* (1) strongly pointing the enhancer locus being a specific *POLG* regulator (Fig 3A and D). These evidences support EE1-3, with a locally transcribed lncRNA, to be regulatory for *POLG* expression.

LncRNAs are often structural components of EEs when transcribed on site: They mediate chromatin looping and physically link the enhancer and promoter together, thereby inducing target gene transcription (Orom & Shiekhattar, 2011; Plank & Dean, 2014). The joint EE1-3/*LINC00925/Ai854517* locus suggested that the lncRNA might be coexpressed with *POLG*. We found *LINC00925/Ai854517* to be expressed exclusively in neural cells (Fig 3F), and its expression levels correlated significantly with *Polg* expression in different developmental stages in mouse CNS (Fig 3G). No *Ai854517* expression was detected in mouse liver, indicating that liver has distinct regulatory mechanisms for POLG expression (Appendix Fig S2A). Repeated experiments to delete conserved regions of *LINC00925* by

**Figure 1.  Three distant enhancers drive *POLG* expression in the central nervous system.**

A    (Left) Luciferase expression in HEK293 cells driven by deletion constructs of *POLG* proximal promoter. (Right) Luciferase expression with mutated CAAT boxes in *POLG* promoter. The error bars indicate standard deviation in three biological replicates. AU; arbitrary units.

B    *POLG* promoter sequence showing predicted CAAT boxes. Red shows disrupted nucleotides by site-directed mutagenesis in (A).

C    Expression pattern driven by 500-bp *Polg* proximal promoter in E12.5 mouse embryo. *LacZ*-positive cell populations in the developing midbrain (black arrows), dorsal root ganglia (gray arrows), and motoneuron progenitors (arrowhead) of the neural tube. Somites show some expression (white arrow). Sectioning planes indicated by red lines. Scale bars 100 μm.

D    Prediction of enhancers in the genomic loci of mtDNA maintenance genes, 100 kb upstream of the analyzed gene, found in human–mouse and human–rat comparisons. EEL score for individual elements: red bars. Protein-coding genes upstream from mtDNA maintenance genes are shown with black lines under each locus (picture not in scale). *POLG* shows three highly conserved elements in a gene-poor region. *TWNK* shows one distant element, with several genes between the element and the gene, suggesting the element not be a specific regulator for *TWNK*. TWNK, Twinkle mtDNA helicase; *POLG*, DNA polymerase gamma, catalytic subunit; *POLG2*, DNA polymerase gamma, accessory subunit; *SSBP1*, single-stranded DNA-binding protein 1; *TFAM*, mitochondrial transcription factor A.

E–G    *POLG* enhancer elements are functional *in vivo* and drive expression in E12.5 transgenic mice. Sectioning planes indicated by red lines. Black line in (E) marks the expression in the dorsal neural tube, whereas rostral and caudal regions lack dorsal expression (black arrows).

H–J    Neural tube *lacZ* expression driven by (H) EE1: immature neuronal precursors (gray arrow), (I) EE2: dorsal neural tube (gray arrow) and dorsal root ganglia (black arrow), and (J) EE3: dorsal neural tube (gray arrow). Scale bars 100 μm.

K–M    Midbrain *lacZ* expression, driven by (K) EE1, (L) EE2, and (M) EE3. Black arrow indicates neuronal population from EE1 embryo stained in (N). Scale bars 100 μm.

N    EE1 drives expression in oculomotor complex; immunofluorescent costaining with antibodies against ISL1/2 (motoneurons of oculomotor complex; red) and β-Gal (green). *LacZ* staining of the region in (K); black arrow. Scale bars 50 μm.

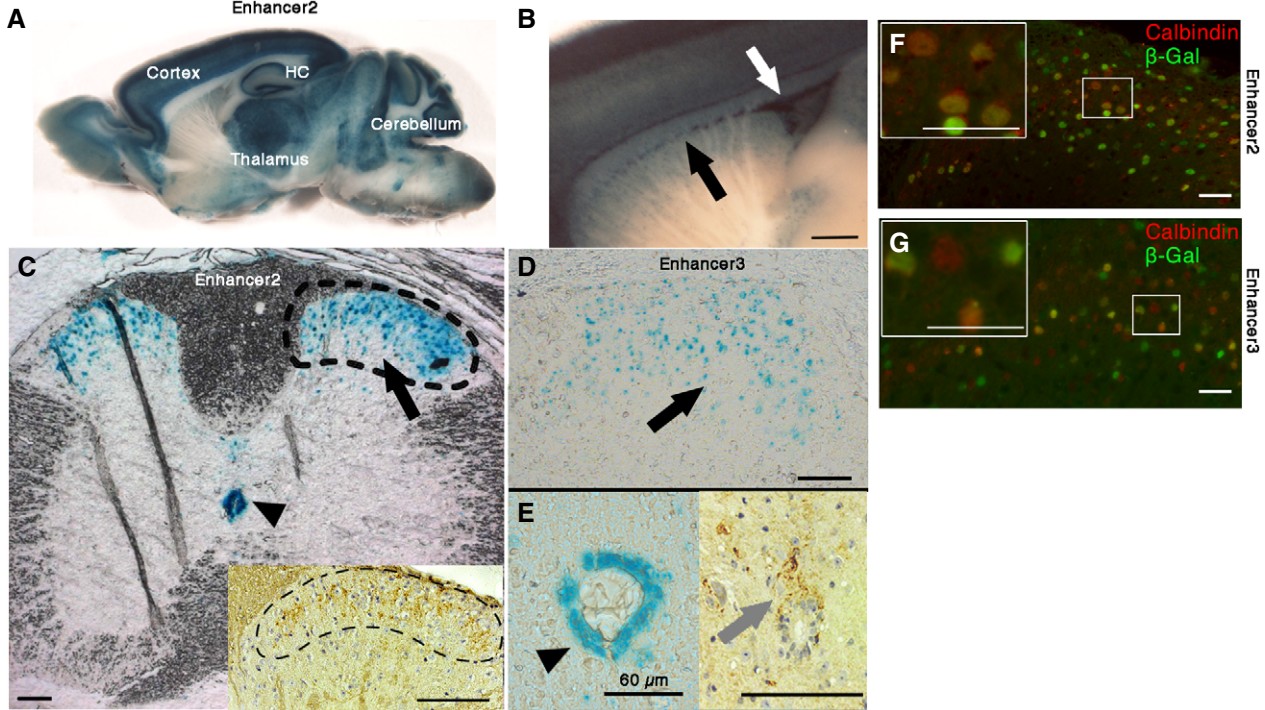

**Figure 2.** *Polg* enhancer EE2 drives expression in adult brain, and EE2 and 3 in sensory interneurons of the adult spinal cord.

A    Sagittal section of adult mouse brain showing *lacZ* expression driven by EE2. HC, hippocampus.

B    EE2-driven expression in rostral migratory stream (black arrow) and in subventricular zone (white arrow). Scale bar 760 μm.

C–G    Adult mouse spinal cord, (C) EE2 and (D, left panel in E) EE3 expression pattern. Dorsal horns, laminae I–III (black arrows) and central canal (arrowheads). *LacZ* staining. Dashed lines in (C) indicate the region of dorsal horn shown in (D), (F), and (G). Insets in (C) and (E) show POLG immunohistochemistry of dorsal horn and central canal (gray arrow), respectively. Scale bars: 100 μm unless indicated otherwise. Calbindin (red) and β-Gal (green) expression in interneurons of the dorsal horn laminae I–III of (F) EE2 and (G) EE3 transgenic adult mice. Immunofluorescence. Scale bars 20 μm.

different Crispr/Cas9 approaches failed to result in targeted clones, which could be a result of growth disadvantage of the targeted clones caused by decreased *POLG* expression and compromised mtDNA replication. *In situ* hybridization showed completely overlapping expression patterns of *Polg* and *Ai854517* transcripts across hippocampus, cortex, and cerebellum (Fig 3H), the areas of EE2 activity in adult mice.

In addition to the EEs, the *LINC00925/Ai854517* intron also harbored the gene for MIR9-3 (Fig 3E). MIR9 is expressed from three different genes, which reside in different genomic regions, have non-overlapping expression patterns but produce identical processed transcripts. Intronic MIRs are commonly transcribed together with the primary transcript (Kim & Kim, 2007), and accordingly, the expression levels of MIR9-3 and *Ai854517* correlated significantly in CNS (Fig 3I). The programs TargetScan, PicTar, miRDB, and PITA predicted six common top targets for MIR9, and MIR9 has recently been shown to downregulate the expression of transcription factors ONECUT1, ONECUT2, and NR2E1 (Madelaine *et al*, 2017). Therefore, we selected these nine putative targets for further analysis (Fig 3J). Overexpression of MIR9 in HEK293 cells significantly downregulated the mRNA expression of NR2E1 and LDLRAP1, and MTHFD2 trended downwards (Fig 3K, Appendix Fig S2B). MTHFD2 was especially interesting, as it is the rate-limiting enzyme of the mitochondrial folate cycle and highly induced as part of the integrated mitochondrial disease stress response in mtDNA

maintenance disease (Bao *et al*, 2016; Nikkanen *et al*, 2016) and a reported target of MIR9-3 (Fig 3K) (Selcuklu *et al*, 2012). Low-density lipoprotein receptor adaptor protein-1 (LDLRAP) showed overlapping expression pattern with EE2/POLG/lncRNA (Lein *et al*, 2007) and has been reported to contribute to lipoprotein internalization for cholesterol synthesis (Mameza *et al*, 2007). NR2E1 is a transcription factor which has been shown to be essential for maintaining neural stem cells in undifferentiated state linking the stem cell pool maintenance to *POLG* expression (Shi *et al*, 2004). We propose that the targets of MIR9 are downregulated specifically in the EE/lncRNA/MIR9/POLG-positive neural cells that promote mtDNA maintenance through *POLG* expression. Furthermore, the results propose an inverse coregulation of *POLG* and *MTHFD2,* with important potential consequences for disease: (i) the activation of EE/lncRNA/MIR9/*POLG* locus and consequent MIR9-mediated suppression of MTHFD2 would blunt the ability of these specific neurons to induce a stress response as a consequence of mtDNA maintenance defect, and that (ii) the same neurons would become especially dependent on CNS folate availability, which is highly regulated through active transport by FOLR1 (Steinfeld *et al*, 2009).

The typical symptoms of POLG patients include sensory neuropathy, ataxia, and progressive ophthalmoplegia, arising from the same regions of CNS where we found EE2 activity: sensory tract of medulla, cerebellum, and oculomotor nucleus. POLG patients often show severe loss of vibration sense and hyperalgesia (Luoma *et al*, 2004;

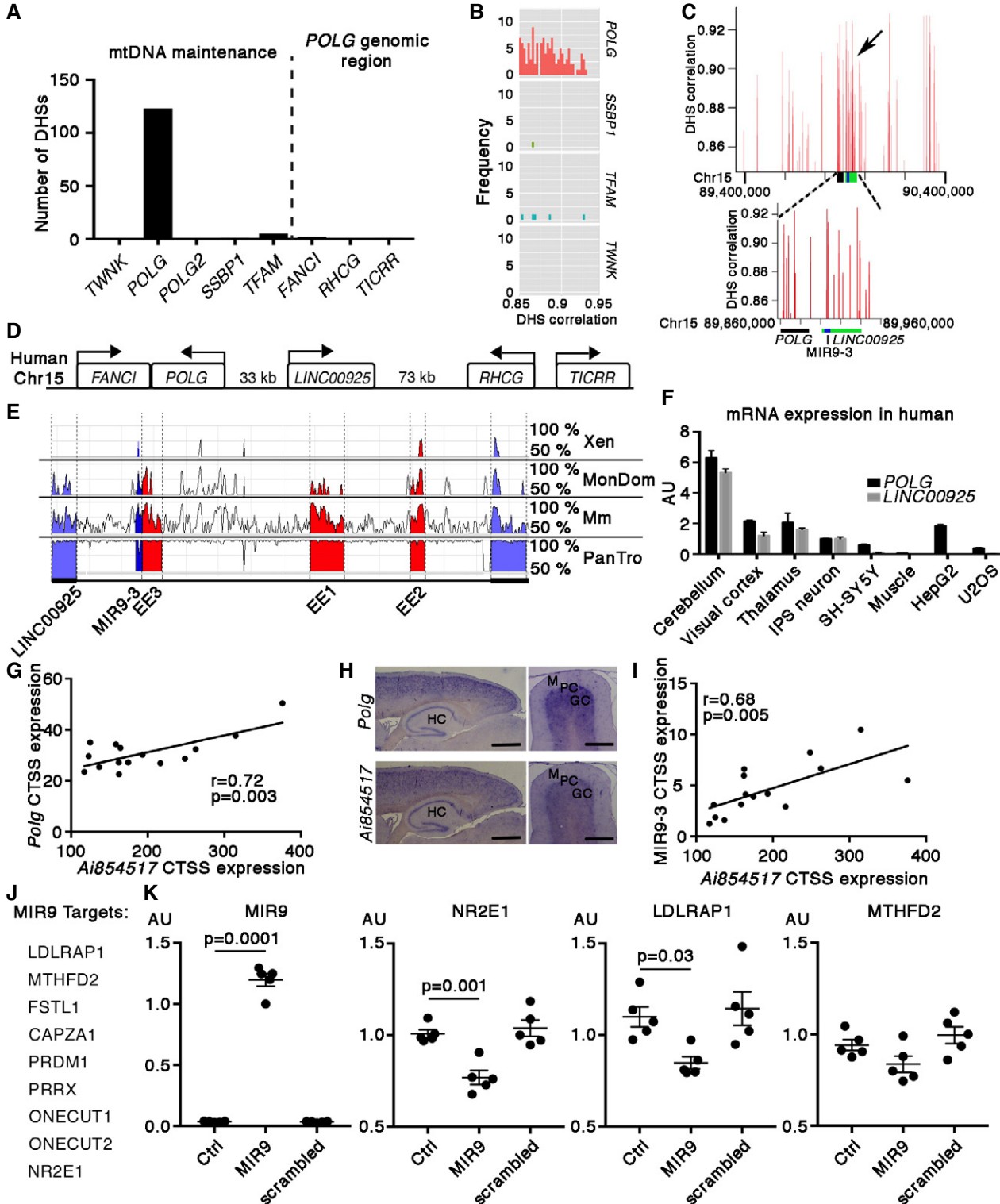

**Figure 3.**

Van Goethem *et al*, 2004; Winterthun *et al*, 2005), as a consequence of damage in the dorsal columns of the spinal cord (Lax *et al*, 2012; Palin *et al*, 2012). We demonstrate here severe degeneration of the dorsal columns of the spinal cord, with preservation of motoneurons of ventral horns, as well as spongiotic degeneration and loss of

neurons in the oculomotor complex in autopsy samples of an adult POLG patient, who manifested with severe sensory neuropathy, ataxia, and ocular muscle paralysis (Fig 4A and B). The muscular versus oculomotor nuclear origin of the extraocular muscle weakness—a typical manifestation of adult-onset mitochondrial diseases—has

**Figure 3.  *LINC00925* regulates *POLG* expression.**

A   The number of distal DNase I hypersensitive sites (DHSs) of mtDNA maintenance genes and genes in the genomic *POLG* locus correlating > 0.85 with target promoter DHS across 125 cell lines. *TWNK*, Twinkle protein; *POLG*, DNA polymerase gamma, catalytic subunit; *POLG2*, DNA polymerase gamma, accessory subunit; *SSBP1*, single-stranded DNA-binding protein 1; *TFAM*, mitochondrial transcription factor A; *FANCI*, Fanconi anemia group I protein; *RHCG*, ammonium transporter Rh type C; *TICRR*, Treslin.

B   Distribution histogram of DHSs for mtDNA maintenance genes.

C   Genomic distribution of *POLG* DHSs. Black arrow shows a cluster upstream from *POLG* coding region.

D   Genes surrounding *LINC00925*.

E   Conservation of regulatory elements of POLG genomic locus in species: *LINC00925*, *POLG* EEs, MIR9-3. PanTro, *Pan troglodytes*; Mm, *Mus musculus*; MonDom, *Monodelphis domestica*; Xen, *Xenopus levis*. Adapted from https://rvista.dcode.org.

F   Expression of *LINC00925* and *POLG* in different human tissues and cell types. Quantitative PCR amplification of cDNA. iPS, induced pluripotent stem cell; SH5Y, neuroblastoma line; HepG2, liver hepatocellular carcinoma line; U2OS, bone osteosarcoma line. Error bars indicate standard deviation in three technical replicates. AU; arbitrary units.

G   *Polg* and long non-coding RNA *Ai854517* (mouse homolog of human *LINC00925*) correlate tightly in mouse cerebellar development. Time points: E18, postnatal days 0, 3, 6, 9; three mice per time point. Expression calculated as cap analysis of gene expression (CAGE) hits in the transcription start site (CTSS).

H   *Ai854517* and *Polg* transcripts colocalize in adult mouse brain; *in situ* hybridization. HC, hippocampus; M, cerebellar molecular layer; PC, Purkinje cell layer; GC, granular cell layer. Scale bars: hippocampus 760 μm, cerebellum 300 μm.

I   Expression levels of MIR9-3 and *Ai854517* correlate in mouse cerebellar development. Time points: E18, postnatal days 0, 3, 6, 9; three mice per time point. Expression calculated as cap analysis of gene expression (CAGE) hits in the transcription start site (CTSS).

J   Predicted targets of MIR9. Six targets were predicted by all prediction programs, TargetScan, PicTar, miRDB, and PITA: low-density lipoprotein receptor adaptor protein-1, LDLRAP1; methylene tetrahydrofolate dehydrogenase-2, MTHFD2; follistatin-like 1, FSTL1; capping actin protein of muscle Z-line alpha-subunit 1, CAPZA1; PR/SET domain 1, PRDM1; paired related homeobox 1, PRRX1. One cut homeobox 1 and 2 (ONECUT1 and ONECUT2) and nuclear receptor subfamily 2 group E member 1 (NR2E1) are recently discovered MIR9 targets.

K   RNA expressions of MIR9, *NR2E2*, *LDLDRAP1*, and *MTHFD2* in HEK293 untransfected controls and in cells transfected with pre-MIR9 or scrambled RNA. Shown is mean with standard error of the mean of 5 analyzed replicates. Statistical testing was performed using one-way ANOVA with Dunnett's correction for multiple comparisons. AU; arbitrary units.

been debated already in 1990s (Rowland *et al*, 1997), and is clinically challenging to assess. Our finding of strong EE/lncRNA/MIR9-regulated POLG expression in oculomotor nucleus already early in embryogenesis argues for special importance of mtDNA maintenance in those neurons, and suggests that POLG disease-associated ophthalmoplegia may be of central origin. Overall, our findings indicate remarkable overlap of the temporally and regionally driven POLG expression and the CNS regions degenerating in POLG disease.

We report that mtDNA maintenance is regulated by a non-coding genomic complex driving *POLG* expression in specific neuronal populations. The functional connection of the enhancer *LINC00925* locus on *POLG* transcription is supported by (i) coactivation of the enhancer MIR9/*LINC00925* locus and *POLG*, demonstrated by an unsupervised DHS analysis; (ii) significant correlation of regulatory transcripts and POLG expression; (iii) specific degeneration of the enhancer-active regions in patients with POLG disorders showing

that these regions are susceptible to POLG defects. These data have high importance in understanding the possible origin of the symptoms of POLG disorders and direct the research of POLG disease mechanisms toward more detailed neuronal populations. Moreover, our study suggests that the *POLG* regulatory locus has potential to modify distant genes and functions through MIR9 regulation. The in-trans link to NR2E1, involved in stem cell maintenance, is interesting, as POLG is known to affect somatic stem cells in nervous system: mtDNA mutator mice with exonuclease-deficient POLG manifest respiratory chain deficiency especially in the stem cell region of subventricular zone, and show reduced stemness properties (Ahlqvist *et al*, 2012). Furthermore, the in-trans link between POLG and mitochondrial one-carbon metabolism would make the specific neurons expressing POLG in high levels especially vulnerable to decrease in CNS folate levels. The data suggest folate to be a modifier of POLG disease manifestations. Our evidence indicates

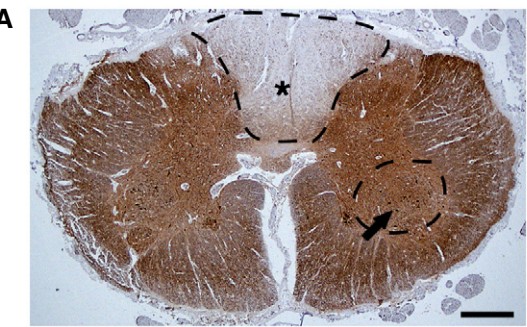
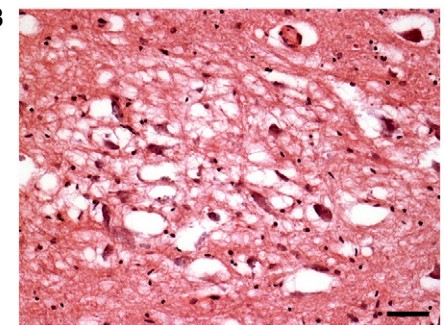

**Figure 4.   POLG defects cause degeneration of enhancer-active regions of CNS.**

A   Spinal cord (C8 level) of MIRAS patient, neurofilament staining. Dorsal columns, especially the gracile, show pallor (dashed line with star); motoneurons in the anterior horns are preserved (dashed line with black arrow). Scale bar 1 mm.

B   Oculomotor complex of POLG patient with progressive ophthalmoplegia; midbrain level. Hematoxylin–eosin staining. The whole area shows spongiotic degeneration. Scale bar 50 μm.

    

that housekeeping proteins, such as the mitochondrial replicase required in all cells with mtDNA replication, may have genomic tissue-specific regulators that carry potential to modulate cell-specific sensitivity to secondary stresses/functions and thereby susceptibility to disease.

# Materials and Methods

### Prediction of enhancer elements

Enhancer Element Locator version 1.5.2.2 was used in enhancer element predictions. The coding regions and 100 kb upstream and downstream of the target genes were analyzed in human, mouse, and rat. The three *POLG* elements characterized in the article were found in human–mouse, human–rat, and mouse–rat searches. All transcription factor matrices provided by the program together with added matrices for AHRARNT.01 and NFX_ARNT.01 (Genomatix) were used in the analysis.

### Prediction of transcription factor binding sites

*POLG* proximal promoter was analyzed for transcription factor binding sites with Genomatix Matinspector (www.genomatix.de).

### Luciferase assay

Luciferase assays were done by transfecting HEK293 with 12.5 ng of promoter constructs (firefly luciferase) and 7 ng of Renilla luciferase as control. Transfections were done by using Lipofectamine 2000 following the instruction of the manufacturer. Dual-luciferase reporter assay system (Promega, #E1910) was used in determining the luciferase activity 24 h after the transfection by following instructions of the manufacturer. Firefly luciferase activity was normalized by Renilla luciferase activity.

### MIR9 overexpression

Pre-MIR9 and scrambled control were transfected into HEK293 cells by Neon electroporation system (Thermo Fisher). Electroporation conditions: 1,100 V, 20 ms, 2 pulses.

### Animal experiments

Animal work was approved by the ethical board of state province office for animal experimentation of Finland, and experiments were conducted according to the guidelines of the same ethical board. C57BL/6 male mice (age: 8–24 weeks) were used in the experiments. Mice were housed at 22°C with 12-h light/12-h dark cycles and had access to standard laboratory chow and water *ad libitum*. Sacrifice was performed by $CO_2$ asphyxia followed by cervical dislocation. Before cardiac perfusion, animals were anesthetized by intraperitoneal injections of pentobarbital sodium (dose: 60 mg/kg).

The mouse enhancer elements and 250- to 300-bp flanking regions were cloned into tkPD vector (gift from Jussi Taipale) containing *lacZ* marker gene and minimal HSV-tk promoter. *LacZ* expression patterns shown were detected at least in three

independent founder lines. The vector for *in vivo Polg* proximal promoter studies was constructed by first removing the HSV-tk promoter from the tkPD vector by BamHI digestion and replacing it with *Polg* proximal promoter. The primer sequences used for cloning are provided in Appendix Table S2.

Before pronuclear injections, the constructs were linearized and prepared according to the instructions of the transgenic mouse unit.

### Human samples

The human samples were obtained after informed consent according to WMA Declaration of Helsinki and the Department of Health and Human Services Belmont Report. The experiments were approved by the Ethical Review Board of Helsinki University Hospital.

### Site-directed mutagenesis

Two nucleotide changes were introduced to the promoter constructs by site-directed mutagenesis to disrupt the CAAT boxes. PCR primers introduced CA->TG change in the core nucleotides of the binding site.

### *LacZ* staining

Cardiac perfusion was performed to adult mice with ice-cold PBS before tissue dissection. Tissues and freshly collected E12.5 mouse embryos were stained for *lacZ* expression using the following protocol. Samples were washed once with 0.1 M phosphate buffer, pH 7.3, and fixed for 30 min in fixing solution (0.2% glutaraldehyde, 5 mM EGTA, 2 mM $MgCl_2$ in 0.1 M phosphate buffer) at room temperature. Next, embryos were washed 3 × 15 min with washing buffer (2 mM $MgCl_2$, 0.01% DOC, 0.02% Nonidet P-40, 0.1 M phosphate buffer). Staining was done in X-Gal staining solution (5 mM K-Ferrocyanide, 2.5 mM K-Ferricyanide, 1 mg/ml X-gal in washing buffer) at 37°C upon agitation in dark. The intensity of the staining was monitored and stopped when desired intensity was reached (3 h–overnight). The stained embryos or tissues were rinsed and washed overnight at +4°C with washing buffer and fixed with 4% PFA overnight at 4°C.

### Immunofluorescence staining

Cardiac perfusion with 4% PFA was performed to all mice before tissue embedding into paraffin. The primary antibodies for immunofluorescent stainings: anti-ISL1/2 (gift from Juha Partanen) 1:200, anti-calbindin (AB11426) 1:200, and anti-β-Gal (AB9361) 1:1,000. Secondary antibodies with appropriate fluorescent dye were all used 1:200 (Invitrogen). Sections were covered with Vectashield mounting medium (Vector Laboratories).

### *In situ* hybridization

Non-radioactive mRNA *in situ* hybridization (ISH) was carried out as described (Copp & Cockroft, 1990). Digoxigenin (DIG)-labeled antisense cRNA probes were transcribed from plasmids according to standard protocols. Primer sequences for *in situ* probes are provided in Appendix Table S2.

**The paper explained**

**Problem**
DNA polymerase gamma (POLG) is a common cause for mitochondrial disorders causing neurodegeneration. POLG patients commonly manifest with sensory neuropathy, epilepsy, and ataxia. The underlying causes for the nervous system-specific manifestations of POLG diseases are currently not understood. Moreover, genomic regulation of nuclear-encoded mitochondrial gene expression as a contributor to tissue specificity remains unexplored.

**Results**
We identified a genomic regulatory locus that drives POLG expression to central nervous system. This locus functions through three enhancer elements but also expresses two functional RNAs, *LINC00925* and MIR9. The targets of MIR9 include NR2E1 and mitochondrial folate cycle linking POLG expression in the specific neurons in trans to neural stem cell maintenance and vitamin B9 metabolism. The enhancer elements drive POLG expression to specific neuronal populations, including the oculomotor nucleus and sensory interneurons of the spinal cord, which we also found to degenerate in POLG patients.

**Impact**
Our results raise an important, previously unconsidered genetic contributor to mitochondrial disease tissue specificity, namely distant regulatory enhancer locus with non-coding RNAs coregulated with the disease gene. The remarkable overlap between the neuronal regions with *POLG* enhancer activity and those that degenerate in patients strongly suggests that the MIR9 targets sensitize the specific neurons to death. One of the targets is mitochondrial folate cycle, recently implicated in mtDNA maintenance disorders, adding a direct link between *POLG* locus and folate metabolism, and suggesting that the coregulation may underlie nervous system manifestations of POLG patients.

## Immunohistochemistry staining

After standard deparaffination, antigen retrieval was performed by 5-min incubation in 1% SDS in TBST followed by 3 × 5-min washes in TBST and 30-min incubation in 2–10 μg/ml (0.01%) saponin in TBST. POLG (Santa Cruz, sc-5930) antibody dilution was 1:50 in 2% BSA. Secondary antibody treatment and staining were done by using KPL HistoMARK (71-00-26) kit.

## Quantitative real-time PCR

Reverse transcription was performed with 500 ng of total RNA using Maxima First Strand cDNA Synthesis Kit for RT–qPCR (Thermo Fisher). The qRT–PCR was performed using DyNAmo Flash SYBR Green qPCR Kit on Bio-Rad CFX96 Real-Time System. *ACTB* and *Actb* were used as reference genes for human and mouse targets, respectively. Primer sequences are provided in Appendix Table S2. MicroRNA amplification was performed using LNA primers for MIR9 (Exiqon, #204513), and SNORD49A (Exiqon, #203904) was used as a reference gene.

## DHS analysis

Public data deposition from DNase I-seq experiments was utilized in identifying the regulatory distal DHSs for mtDNA maintenance genes as in Ref. (Thurman *et al*, 2012).

## Mouse cerebellum expression analysis

Fantom5 database was utilized for expression analysis of *Polg*, *Ai854517*, and MIR9-3 in mouse cerebellum (Abugessaisa *et al*, 2016). Sample IDs: 10145-102I1–10145-102I5.

## Prediction of MIR9 targets

A gene was considered a likely target of MIR9, if it was included in the top 300 predicted targets by TargetScan, PicTar, miRDB, and PITA prediction programs.

**Expanded View** for this article is available online.

## Acknowledgements

The authors wish to thank Anu Harju for experimental expertise, Timo Otonkoski for technical consultations, and the following funding sources: European Research Council, Academy of Finland, Sigrid Jusélius Foundation, Jane and Aatos Erkko Foundation, University of Finland (for A.S.), Helsinki Biomedical Graduate School (for J.N.).

## Author contributions

JN designed the study, performed experiments, analyzed and interpreted results, and wrote the manuscript; JCL, DB, and MH performed experiments and analyzed and interpreted results; JP supervised the study and analyzed data; AP performed patient autopsy and analyzed data; PI collected patient samples and clinical data, and together with VB analyzed and interpreted data; AS designed the study, supervised experiments, interpreted data, and wrote the manuscript. All authors commented on the manuscript.

## Conflict of interest

The authors declare that they have no conflict of interest.

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
