## [Review Process File · EMBO Molecular Medicine]

A Complex genomic locus drives mtDNA replicase POLG expression to its Disease-Related Nervous System Regions

Joni Nikkanen, Juan Cruz Landoni, Diego Balboa, Maarja Haugas, Juha Partanen, Anders Paetau, Pirjo Isohanni, Virginia Brilhante, Anu Suomalainen

Corresponding author: Anu Suomalainen, University of Helsinki

Review timeline:

Submission date:	05 May 2017
Editorial Decision:	10 June 2017
Revision received:	18 September 2017
Editorial Decision:	29 September 2017
Revision received:	03 October 2017
Accepted:	11 October 2017

Transaction Report:

Editor: Roberto Buccione & Céline Carret

1st Editorial Decision

10 June 2017

Thank you for the submission of your manuscript to EMBO Molecular Medicine and many apologies for the unusual delay in providing you with a decision.

We have now received comments from the two out of the three Reviewers whom we asked to evaluate your manuscript. We have been unable so far, to retrieve the third.

Hence, to avoid further delays I am sending the two consistent evaluations of Reviewers 1 and 2 at this time. I will forward Reviewer 3's delayed report, if and as soon as we are able to obtain it. When (within reason) this report does arrive and if it raises additional important issues that have to be addressed to support this study, these would also need to be taken into consideration in your revision. Please note that I would not ask you to consider further-reaching requests with respect to the current evaluations.

You will see that in aggregate, both reviewers find the study of certain interest, while at the same time clearly mentioning the need to clarify a number of issues. I will not dwell into much detail, as the comments are self-explanatory. However, I would like to specifically point out that both reviewers note the lack of functional correlation, and therefore the case for translational relevance, regarding the failed attempt to target LINC00925. This aspect, as you know, is especially relevant for EMBO Molecular Medicine and therefore, I must ask you to improve this aspect by providing supporting evidence beyond the plausible correlation. I note that reviewer 1 suggests some possible strategies to address the issue.

In conclusion, while publication of the paper cannot be considered at this stage, we would be

pleased to consider a suitably revised submission, provided, however, that the Reviewers' concerns are fully addressed with further experimentation where required.

Please note that it is EMBO Molecular Medicine policy to allow a single round of revision only and that, therefore, acceptance or rejection of the manuscript will depend on the completeness of your responses included in the next, final version of the manuscript.

As you know, EMBO Molecular Medicine has a "scooping protection" policy, whereby similar findings that are published by others during review or revision are not a criterion for rejection. However, I do ask you to get in touch with us after three months if you have not completed your revision, to update us on the status. Please also contact us as soon as possible if similar work is published elsewhere.

I look forward to seeing a revised form of your manuscript as soon as possible.

***** Reviewer's comments *****

Referee #1 (Remarks):

Nikkanen J et al, report the identification of a complex genomic regulatory locus that could account for the central nervous system tissue (CNS) specificity in POLG-related diseases. Novelty and originality of the research study in the mtDNA maintenance field are with no doubt. However, the conclusions are not fully supported by the results as detailed below:

1) The identified enhancers clearly localized in the area of degeneration in patients with POLG disease. However, the direct link between physiology and pathology is missing. The authors tried to delete the conserved regions of LINC00925 by CRISP/CAS9 technology but they failed in targeting clones. Did they perform down-regulation of the LINC00925 by siRNA in mouse or human cell line? Is there any response in term of mRNA expression of the control regions in the presence of POLG defect in patients or mouse biological tissues/cells? Is there an increase of POLG mRNA, protein and activity level when overexpressing the enhancer?

2) The authors refer to POLG-disease as mostly CNS disorders and they omit the liver involvement that is definitely present in at least two clinical syndromes associated with POLG gene defect (Alpers-Huttenlocher syndrome and Childhood myocerebrohepatopathy spectrum). Did the authors analyze the enhancer expression in liver tissues of adult mice? Are data available in patients' biological tissues? Did the authors consider the presence of additional loci driving the tissue specificity in liver? A comment on this would be valuable in the manuscript.

3) It is interesting the potential role of folate availability in developing POLG disease. However, a previous study (cit.Nikkanen et al 2016) in the deleter Polg mouse demonstrated that folate supplementation didn't modify the mitochondrial dysfunction in affected tissues. Are additional data such as folate levels in patients biological tissues/cells available for supporting this hypothesis? In the absence of more strongly supportive data, I would suggest to omit the last conclusive statement regarding the folate availability.

Additional comments

Title: It is not clear from the title whatever the authors aim to describe a common process in mtDNA replication defects or a role of the enhancers in the tissue specificity of POLG-related disease. I would suggest modify the title and detail more this aspect in the report.

Figure 2c: It is not very clear the co-localization of lacZ staining and POLG protein immunohistochemistry. I would suggest the use of the same magnification for the two pictures (lacZ staining and POLG immunohistochemistry).

In conclusion, this is an interesting, original and novel study that demonstrates the presence of a complex genomic regulatory region driving the CNS expression of mt-DNA replicase. I believe that the suggested revisions will increase the translational value of the manuscript. Therefore, I will strongly recommend the manuscript for publication after revisions.

Referee #2 (Remarks):

Nikkanen and colleagues have examined the question of what underlies the variability of Polg related disease and whether this could be explained by the presence of tissue specific genomic regulators. They therefore studied the promotor region of POLG and found that it was highly active in specific CNS sites at day E12.5. Thereafter, they looked for enhancer elements using prediction programmes 100kb up/downstream of the POLG gene and found 3. These were separately examined in transgenic mice and all were active at E12.5 in different CNS regions including the oculomotor complex (EE1) superior colliculi (EE2) and midbrain. When they looked in adult mouse brain, they found EE2 was expressed widely but most appeared in hippocampus, cortex, thalamus, external layer of olfactory bulb and both the purkinje and granular cell layers of the cerebellum. EE1 was not seen and EE3 inconclusive. In the spinal cord EE2 & 3 were seen in dorsal horn. DNAase sensitivity mapping was performed looking for coincidental opening of regions distal and proximal to the transcription start site and 123 sites were discovered. The distal sites overlapped with a long non coding RNA Ai854517 and within the intron were EE1-3. Interesting a micro RNA was also found within the same region and in silico analysis suggested targets including LDLRAP and MTHFD2.

Comments to the authors

This is interesting work that begins to provide some understanding of why different regions of the CNS manifest disease or not and how an apparently essential gene such as POLG, can be temporally and regionally controlled. The work appears methodologically and scientifically sound and the conclusions, at least from the direct experimental work, supported by the data. There is a tendency to over speculate about the role of MTHFD2 and mtDNA maintenance, which is based on in silico assumptions of predicted targets for MIR9. It does however, fit with earlier work from this group suggesting a defect in 1 carbon metabolism. The work also provides some intriguing insight in to what drives the ophthalmoplegia.

1. We are shown data from the transgenic mice containing the POLG promotor driving luciferase at day E12.5. It is important to know what the patterns were at other time points. Were there other sites that showed regulated expression at other developmental time points?
2. In a related question, there was evidence that the enhancers (EE2) were also active in the adult brain. While there was increased expression in thalamus and dentate, areas known to be involved in the human disease, there was also increased activity in the hippocampus, which is not. Is this a mouse specific pattern of expression?
3. The work beyond that looking at Polg expression and enhancers is largely conjecture. Interesting as it may be, the data concerning the lncRNA suggests co-expression but does not prove that a functional link exists. The authors tried deleting parts of Ai854517 without success. It would be good to see some functional correlation whether in mice or some other organism. This is similarly true for the MIR9-3 data. evidence of functional

1st Revision - authors' response

18 September 2017

Referee #1 (Remarks):

Nikkanen J et al, report the identification of a complex genomic regulatory locus that could account for the central nervous system tissue (CNS) specificity in POLG-related diseases. Novelty and originality of the research study in the mtDNA maintenance field are with no doubt. However, the conclusions are not fully supported by the results as detailed below:

The authors would like to thank the reviewer for the positive comments and for sharing the enthusiasm for the novel findings of the manuscript.

1) The identified enhancers clearly localized in the area of degeneration in patients with POLG disease. However, the direct link between physiology and pathology is missing. The authors tried to delete the conserved regions of LINC00925 by CRISP/CAS9 technology but they failed in targeting

clones. Did they perform down-regulation of the LINC00925 by siRNA in mouse or human cell line? Is there any response in term of mRNA expression of the control regions in the presence of POLG defect in patients or mouse biological tissues/cells? Is there an increase of POLG mRNA, protein and activity level when overexpressing the enhancer?

We thank the reviewer for thoughtful experimental suggestions. We have considered all these, but the approaches other than full-KO are challenging because the target is a long non-coding RNA, which functions in the nucleus. siRNAs are useful for protein downregulation but their effect in the nucleus is limited. Also, transgenic overexpression of enhancer-associated-lnc (if this is what the reviewer means by overexpressing the enhancer) is not informative because the structural lnc needs to be expressed from its own locus. After transcription, the RNA remains partially attached to its gene locus and interacts with a protein scaffold in the target promoter (Mao et al. 2011, Nat Cell Biol, 13; 95-101). Overexpression from a distant locus would not have the local chromatin function. Since our CRISPR KO clones failed to grow, we have now tuned down the LNC-discussion. We refer to the LINC00925 and POLG connection as ‘coexpression’, which is shown in Figure 3G.

Specific quantitative neuronal RNA-analysis from autopsy material is not reliable from single samples: the neuronal populations of interest are dead and the quality of RNA is heavily influenced by post-mortem degradation and time before sampling in different patients & controls. This is a direction we wish to take but do not have yet enough of autopsy brains to make conclusions, and therefore, the experiments will not be available for article – if ever. Mouse models with human POLG disease mutations, or replicating POLG-disease findings have not been reported. We would like to emphasize, however, that with human materials associations can be reached, and the correlations of the expression of POLG, MIR-9 and LINC00925 are very significant ($p < 0.005$), and conserved in humans and mice. The chromosomal region is open at the same time only in POLG locus –not with other mtDNA replisome proteins – further emphasizing the role of POLG in these specific neurons.

2) The authors refer to POLG-disease as mostly CNS disorders and they omit the liver involvement that is definitely present in at least two clinical syndromes associated with POLG gene defect (Alpers-Huttenlocher syndrome and Childhood myocerebrohepatopathy spectrum). Did the authors analyze the enhancer expression in liver tissues of adult mice? Are data available in patients' biological tissues? Did the authors consider the presence of additional loci driving the tissue specificity in liver? A comment on this would be valuable in the manuscript.

We would like to thank the reviewer for this correct note. Indeed, liver is often affected in POLG diseases especially when the brain is involved, and not emphasizing it was an oversight because the focus in the paper is in CNS enhancers. We have now included the liver involvement to the introduction: “POLG is a prime example of tissue- and genotype-specific variability of manifestations, and POLG mutations cause highly tissue-specific disorders typically manifesting in the nervous system but also affecting liver and skeletal muscle.” and “The clinical manifestations of the mutations of POLG are especially diverse: 1) Alpers-Huttenlocher syndrome, an epileptic encephalohepatopathy of early childhood, manifesting during the first years of life...”

We also analyzed the lincRNA expression by QPCR in adult mouse liver as the reviewer suggested. Low or no expression was detected in liver whereas cerebellum showed robust expression. The data support further the CNS-specificity of the regulatory system. The following sentence is now added to the results and the data is illustrated in Appendix Figure S3A: “No *Ai854517* expression was detected in mouse liver indicating that liver has distinct regulatory mechanisms for POLG expression (Fig.S3a).”

*3) It is interesting the potential role of folate availability in developing POLG disease. However, a previous study (cit.Nikkanen et al 2016) in the deleter *Polg* mouse demonstrated that folate supplementation didn't modify the mitochondrial dysfunction in affected tissues. Are additional data such as folate levels in patients biological tissues/cells available for supporting this hypothesis? In the absence of more strongly supportive data, I would suggest to omit the last conclusive statement regarding the folate availability.*

We would like to correct this statement, since the Deletor mouse the reviewer is referring to, is a transgenic mouse carrying a *TWNK* patient mutation, not *POLG*, and is a model for a muscle-manifesting disorder (also patients with the same mutation manifest only in the muscle). Therefore, it is not directly relevant for this question. However, we have now omitted the last sentence from the manuscript.

Additional comments

Title: It is not clear from the title whatever the authors aim to describe a common process in mtDNA replication defects or a role of the enhancers in the tissue specificity of POLG-related disease. I would suggest modify the title and detail more this aspect in the report.

The point of the Reviewer is good, to include the disease relevance to the heading. We have now modified the heading as follows: “Complex Genomic Locus Drives mtDNA replicase *POLG* to its Disease-Related Nervous System Regions”

Figure 2c: It is not very clear the co-localization of lacZ staining and POLG protein immunohistochemistry. I would suggest the use of the same magnification for the two pictures (lacZ staining and POLG immunohistochemistry).

We have now corrected this and provide a picture with higher magnification which is incorporated in the figure.

In conclusion, this is an interesting, original and novel study that demonstrates the presence of a complex genomic regulatory region driving the CNS expression of mt-DNA replicase. I believe that the suggested revisions will increase the translational value of the manuscript. Therefore, I will strongly recommend the manuscript for publication after revisions.

Referee #2 (Remarks):

Nikkanen and colleagues have examined the question of what underlies the variability of Polg related disease and whether this could be explained by the presence of tissue specific genomic regulators. They therefore studied the promotor region of POLG and found that it was highly active in specific CNS sites at day E12.5. Thereafter, they looked for enhancer elements using prediction programmes 100kb up/downstream of the POLG gene and found 3. These were separately examined in transgenic mice and all were active at E12.5 in different CNS regions including the oculomotor complex (EE1) superior colliculi (EE2) and midbrain. When they looked in adult mouse brain, they found EE2 was expressed widely but most appeared in hippocampus, cortex, thalamus, external layer of olfactory bulb and both the purkinje and granular cell layers of the cerebellum. EE1 was not seen and EE3 inconclusive. In the spinal cord EE2 & 3 were seen in dorsal horn. DNAase sensitivity mapping was performed looking for coincidental opening of regions distal and proximal to the transcription start site and 123 sites were discovered. The distal sites overlapped with a long non coding RNA Ai854517 and within the intron were EE1-3. Interesting a micro RNA was also found within the same region and in silico analysis suggested targets including LDLRAP and MTHFD2.

Comments to the authors

This is interesting work that begins to provide some understanding of why different regions of the CNS manifest disease or not and how an apparently essential gene such as POLG, can be temporally and regionally controlled. The work appears methodologically and scientifically sound and the conclusions, at least from the direct experimental work, supported by the data. There is a tendency to over speculate about the role of MTHFD2 and mtDNA maintenance, which is based on in silico assumptions of predicted targets for MIR9. It does however, fit with earlier work from this group suggesting a defect in 1 carbon metabolism. The work also provides some intriguing insight in to what drives the ophthalmoplegia.

We would like to thank the Reviewer for his/her positive comments, with special notes of importance to clinical manifestations.

1. We are shown data from the transgenic mice containing the POLG promotor driving luciferase at

day E12.5. It is important to know what the patterns were at other time points. Were there other sites that showed regulated expression at other developmental time points?

We do agree with the reviewer that studying developmental roles of POLG proximal promoter would be interesting. However, we have focused on the characterization of the novel regulatory locus in adult life since POLG patients do not show developmental problems. Why not, is an interesting question but not in the scope of this article.

2. In a related question, there was evidence that the enhancers (EE2) were also active in the adult brain. While there was increased expression in thalamus and dentate, areas known to be involved in the human disease, there was also increased activity in the hippocampus, which is not. Is this a mouse specific pattern of expression?

Severe epilepsy is a key manifestation of the POLG diseases, and hippocampus is a key region involved in epilepsy (overview for example Ang CW et al. Massive and specific dysregulation of direct cortical input to the hippocampus in temporal lobe epilepsy. J Neurosci 2006). With this note, we would also like to emphasize that the overlap of POLG disease-affected regions and the enhancer-active regions are intriguingly overlapping. However, we do not claim to explain all the patient manifestations with the enhancers; their importance is the existence of such regulatory loci, which have potential to modulate neuronal susceptibility to mitochondrial disease progression – for example through folate availability. The main point of the paper is that even constitutively required proteins, such as POLG needed for mtDNA replication presumably in all cells, may have genomic tissue-specific regulators, that have potential to modulate tissue-specific sensitivity to secondary stresses / functions. This has now been further clarified in the discussion, p.9-10.

3. The work beyond that looking at Polg expression and enhancers is largely conjecture. Interesting as it may be, the data concerning the lncRNA suggests co-expression but does not prove that a functional link exists. The authors tried deleting parts of Ai854517 without success. It would be good to see some functional correlation whether in mice or some other organism. This is similarly true for the MIR9-3 data.

We agree with the reviewer that the expression of LINC00925 and POLG is correlative (but strong correlation with $r=0.72$). Therefore, we now refer to the connection as ‘coexpression’ which is shown in Figure 3G. We think that the expression of the LINC00925 and MIR9 in the same cells where POLG is expressed is an important observation and raise a possibility of a completely novel *in trans* connection of POLG and MIR9 targets, which function in cellular metabolism beyond mitochondrial ATP production.

We have now performed cell culture experiments to provide mechanistic experimental data for MIR9 as the reviewer suggested. By overexpressing MIR9 in HEK293 cells we verified the biological targets of the MIR9 and found in addition to original data of MTHFD2, also interesting regulatory links to stem cell maintenance, which is known to be affected in POLG-progeria mice, the mtDNA Mutators (Ahlqvist et al, 2012 Cell Metab). These data are now shown in Figure 3K and Appendix Figure S3B.

2nd Editorial Decision

29 September 2017

Thank you for the submission of your manuscript to EMBO Molecular Medicine. We have now heard back from the two Reviewers whom we asked to evaluate your manuscript.

As you will see the Reviewers are now satisfied with your manuscript and we are thus prepared to accept it for publication pending editorial requests:

1) As per our Author Guidelines, the description of all reported data that includes statistical testing must state the name of the statistical test used to generate error bars and P values, the number (n) of independent experiments underlying each data point (not replicate measures of one sample), and the actual P value for each test (not merely 'significant' or ' $P < 0.05$ ').

2) All animal experimentation must be fully detailed in the Materials and Methods section. I notice a

lack of information in this respect (in vivo expression, cardiac perfusion, euthanasia, etc).

Please submit your revised manuscript within two weeks.

***** Reviewer's comments *****

Referee #1 (Remarks for Author):

The Authors have carefully considered all of the suggested revisions and potential criticism to the results and their interpretation. Accordingly to this, the manuscript has been accurately revised and, in my opinion, it's suitable for publication.

Referee #2 (Remarks for Author):

I agree that there is sufficient evidence to suggest that tissue specific regulation of POLG is present. I think that the authors have addressed most of my concerns raised and what remains is sufficiently interesting to warrant placing the data in the public domain.

Corresponding Author Name: Anu Suomalainen

Manuscript Number: EMM-2017-07993-V3